# The Potential Roles of Extracellular Vesicles as Biomarkers for Parkinson’s Disease: A Systematic Review

**DOI:** 10.3390/ijms231911508

**Published:** 2022-09-29

**Authors:** Jessica Valencia, Marta Ferreira, J. Francisco Merino-Torres, Antonio Marcilla, Jose M. Soriano

**Affiliations:** 1Food & Health Lab, Institute of Materials Science, University of Valencia, 46980 Paterna, Valencia, Spain; 2Joint Research Unit on Endocrinology, Nutrition and Clinical Dietetics, Health Research Institute La Fe-University of Valencia, 46026 Valencia, Valencia, Spain; 3Department of Endocrinology and Nutrition, University and Polytechnic Hospital La Fe, 46026 Valencia, Valencia, Spain; 4Department of Pharmacy and Pharmaceutic Technology and Parasitology, University of Valencia, 46010 Burjassot, Valencia, Spain

**Keywords:** Parkinson’s disease, extracellular vesicles, exosomes, biomarkers, α-synuclein, lncRNAs, miRNAs

## Abstract

Parkinson’s disease (PD) is a slowly progressive neurodegenerative disorder, characterized by the misfolding and aggregation of α-synuclein (α-syn) into Lewy bodies and the degeneration of dopaminergic neurons in the substantia nigra pars compacta. The urge for an early diagnosis biomarker comes from the fact that clinical manifestations of PD are estimated to appear once the substantia nigra has deteriorated and there has been a reduction of the dopamine levels from the striatum. Nowadays, extracellular vesicles (EVs) play an important role in the pathogenesis of neuro-degenerative diseases as PD. A systematic review dated August 2022 was carried out with the Preferred Reporting Items for Systematic Reviews and Meta-Analyses with the aim to analyze the potential role of EVs as biomarkers for PD. From a total of 610 articles retrieved, 29 were eligible. This review discusses the role of EVs biochemistry and their cargo proteins, such as α-syn and DJ-1 among others, detected by a proteomic analysis as well as miRNAs and lncRNAs, as potential biomarkers that can be used to create standardized protocols for early PD diagnosis as well as to evaluate disease severity and progression.

## 1. Introduction

Parkinson’s disease (PD) is a slowly progressive neurodegenerative disorder, characterized by the misfolding and aggregation of α-synuclein (α-syn) into Lewy bodies and Lewy neurites and the degeneration of dopaminergic neurons in the substantia nigra pars compacta [1,2]. α-syn, the main fibrillar component of Lewy bodies, is also a protein found in the nervous system of disease-free individuals [3].

Clinical manifestations of PD are estimated to appear once the substantia nigra has deteriorated with a reduction of the dopamine levels from the striatum [4]. Clinical features include motor symptoms, such as resting tremor, muscle tone rigidity, postural and gait instability and bradykinesia [5,6]. As well, nonmotor symptoms are present, such as sleep disorders, cognitive impairment, constipation, dysphagia, salivation, hallucinations and delusions, among others [1,5].

Extracellular vesicles (EVs), as secreted membrane particles that are naturally released by all types of cells [7,8], have been of interest in different fields. They can be present in body fluids such as plasma, urine, milk, saliva, cerebrospinal fluid (CSF), bronchi alveolar lavage and synovial fluid among others [8,9,10]. Perhaps, the main role of EVs is their capacity to mediate cell-to-cell communication by transferring RNA, DNA, signaling complexes, lipids, membrane proteins, etc., in physiological but also pathophysiological conditions [2,11].

Nowadays, terms such as “microvesicles” or “ectosomes” [12], among others, have fallen into disuse, preferring terms such as “exosomes” or mainly “extracellular vesicles” as recommended by the International Society for Extracellular Vesicles in its latest statements [7].

The current knowledge of EVs in the pathogenesis of PD are starting to be elucidated. Nowadays, several research projects are focused on the study of misfolded proteins such as α-syn, DJ-1, long noncoding RNAs (lncRNAs) (which are transcribed RNA molecules with a length greater than 200 nucleotides that do not encode proteins but are involved in the regulation of gene expression through epigenetic, translation, transcription and post-transcription mechanisms) and miRNAs. In fact, the use of more purified isolation and characterization methods for EVs will help find new biomarkers for an early diagnosis of Parkinson’s disease [7,12,13,14].

The aim of this systematic review was to evaluate the viability of EVs as potential biomarkers for the detection and progression of PD. The analysis of EVs derived from different bodily fluids, their isolation methods and distinct cargos, bring us closer to understanding their implication and use as a potential tool for winning a part of the battle in this neurodegenerative disease.

## 2. Methods

This systematic review was developed according to the Preferred Reporting Items for Systematic Reviews and Meta-Analyses (PRISMA) statement as presented in the flowchart below (Figure 1) and dated August 2022 using PubMed and Embase databases.

The search strategy was designed specifically for each of the search engines used in this review.

For the PubMed search, four different search strategies were developed to get the most accurate results:(exosomes[Title/Abstract]) AND parkinson’s disease[Title/Abstract](exosomes[MeSH Terms]) AND parkinson’s disease[MeSH Terms](extracellular vesicles[Title/Abstract]) AND parkinson’s disease[Title/Abstract](extracellular vesicles[MeSH Terms]) AND parkinson’s disease[MeSH Terms]

For the Embase search, the strategy used was:exosomes:ti,ab,kw AND ‘parkinson disease’:ti,ab,kwextracellular vesicles:ti,ab,kw AND ‘parkinson disease’:ti,ab,kw

For inclusion criteria, only case–control studies were considered. As well, the year of publication was limited for this review. During the initial search, a significant increase in the search trends for this topic was observed in recent years (Figure 2). Thus, the year of publication was limited to the last 6 years with the purpose of analyzing the most noteworthy data available. Full texts in English were required in order to retrieve the information needed for this review: title, authors, year of publication, sample size and characteristics, inclusion/exclusion criteria, biological sample type, method of extraction of biological sample, method of isolation of exosomes and main outcomes.

As exclusion criteria, the reasons were report status as conference abstracts, review articles, case reports, letters, editorials, unpublished data, articles without full texts and articles not available in English.

Two teams of paired reviewers (J.V., M.F., J.M.S., A.M.) with expertise in medical and health evaluations and training in research methodology independently screened titles, abstracts, and full texts for eligibility, assessed generalizability, and collected data from each eligible case reports. Any disagreements were resolved by a third researcher (J.F.M.-T.).

## 3. Results and Discussion

A total number of 610 records were retrieved through database searching. After removing duplicates, 339 records were assessed for eligibility, and 29 articles were finally selected for this qualitative synthesis. Table 1 summarizes the records included and analyzed in this review.

EVs play an imperative role in communication, transfer and delivery of information between the CNS and the circulatory system. Herein, the study of circulating EVs and their meaningful biochemistry and cargo (from prion proteins to miRNAs, LncRNAs) seems to elucidate now even more their role in PD pathogenesis.

As seen in Table 1 of all 29 studies analyzed, 22 of them used blood as a biological sample from which EVs were isolated, a clinical accessible test that still has the disadvantage of not reflecting the actual conditions of the CNS during the pathogenesis of PD [18]. The remaining studies used cerebrospinal fluid (two), urine (two), saliva (two) and blood and cerebrospinal fluid (one). In Table 2, we aimed to summarize the potential biomarkers that were analyzed in this review.

### 3.1. α-syn and Its Derivatives

It is widely accepted that the accumulation of misfolded α-syn in PD has a primary role in the pathogenesis of the disease as the main component of Lewy bodies, along with ubiquitinated proteins, that accumulate in the surviving neurons [43]. α-syn can be found in the presynaptic terminals of neurons and participates in synaptic plasticity and vesicle trafficking [5,43]. It can be transferred neuron-to-neuron, with the capacity to form aggregates in the recipient cells but also propagate in a prion-like manner through the system [5].

The role of EVs in the carrying of α-syn as a possible pathological way of spreading for PD has been analyzed in 12 of the 29 studies. It was observed by Cerri et al. [14], Cao et al. [24] and Xia et al. [38] that the peripheral exosomal levels of α-syn in PD patients were higher than the levels in HC. EVs α-syn oligomers have garnered interest as the primary neurotoxic form of seeding healthy neurons [30].

Mutations in the α-syn-encoding gene SNCA, such as A53T, E46P, A30P, H50Q, G51D and A53E, promote the transformation of α-syn into its most toxic forms, fibrils and oligomers [44]. The oligomeric form of α-syn has been identified as the toxic variety participant in PD [30] and makes it easier for recipient cells to take up α-syn in this conformation rather than when it is in a free form [45,46]. In addition, the exosomal environment promotes α-syn aggregation as well, as a way of transportation for its propagation within the CNS and hence for the neurodegenerative process in PD [45,47,48].

Moreover, oligomeric α-syn has also been observed to be promoted by post-translational modifications (PTM) such as phosphorylation, nitration and dopamine (DA) modification. Particularly in PD, phosphorylation at Ser-129 seems to be the most prevalent PTM form of α-syn found in PD with Lewy bodies. It accelerates the formation of α-syn inclusions, as well as neuronal loss in mice [43,49].

Zhao et al. [18] studied DJ-1 and α-syn in plasma neural-derived exosomes, also using neuronal adhesion molecule L1 (L1CAM), and found significantly higher levels in the PD group than in HC. In addition, matching results, the levels of α-syn in neuronal exosomes in PD were higher compared to those in controls after electroanalyses were performed according to Fu et al. [32]. Moreover, Niu et al. [25] sustained that plasma neuronal exosomal α-syn could be involved in the pathophysiologic process of dopaminergic neurodegeneration in PD, based on the association between higher levels and increased risk of motor progression such as its ability to distinguish between early stage PD patients and HC. The expression of the neuronal adhesion molecule L1 (L1CAM) has been considered to reflect CNS status. Not only does L1 act as a cell surface receptor but it may also play a role as a ligand at a considerable distance from its origin [50]. These might explain the inconsistencies in multiple research projects that measured the same protein and why this might be increased in the analyzed samples.

On the contrary, Stuendl et al. [30] and Si et al. [39] analyzed α-syn-containing exosomes isolated from serum (immune-captured by an antibody directed against the neural L1 cell adhesion molecule L1CAM) and CSF samples, respectively, finding that the levels of exosomal α-syn were lower in PD patients than in HC and it was inverse to the course and severity of the disease.

Furthermore, Rani et al. [36] found that a cut-off value of 3.1 × 10^7^ particles/mL of salivary exosomes distinguished the PD vs. HC group with a sensibility and specificity of 100%; however, a larger cohort is needed to validate these results.

Although multiple studies have quantified the levels of α-syn in its multiple conformations, primarily finding an increased in PD patients vs. healthy control groups [18,25,34], the concentration of salivary exosomes should be explored in larger cohorts, as a type of noninvasive biological sample with a 100% sensibility and specificity in prior results.

In brief, the studies published to date show inconsistencies regarding the potential of α-syn to be used as a biomarker for PD, as exposed above. This may have different explanations, such as the lack of standardized protocols for sample processing, the contamination of the actual sample or the variety of conformations studied, from unfolded monomer to complex fibrils such as oligomers and the use of L1CAM as an antibody for neural-derived EVs [3,51].

### 3.2. PrP^c^, DJ-1, OxiDJ-1 and Tau Protein

Cellular prion protein (PrP^c^), a cell-surface glycoprotein highly expressed in central and peripheral nervous system was found increased in plasma exosomes of PD patients vs. HC and associated with cognitive decline [16]. These results sustain the role of PrP^c^ in the physiopathological process of PD.

DJ-1 gene encodes DJ-1, a chaperone protein whose main function is to inhibit the aggregation of α-syn. It is also involved in the protection of neurons against oxidative stress and cell death [44]. It is worth mentioning that OxiDj-1 can be affected by other clinical pathologies that are undiagnosed at the time of taking the sample. Zhao et al. and Jang et al. demonstrated that plasma neural-derived DJ-1 in exosomes and OxiDJ-1 levels in urine, respectively, were increased in PD patients in comparison to healthy individuals. OxiDJ-1 showed significant differences only when ELISA was performed in comparison with a Western blot assay. Thus, OxiDJ-1 could be studied in larger cohorts as an achievable biomarker for PD detection in a near future [18,40].

Tau is a well-known protein aggregated in a hyperphosphorylated form in Alzheimer’s disease. EVs can mediate the spread of toxic forms of tau, helping it go from cell-to-cell and into different areas of the brain. Shi et al. pointed out the association between the levels of tau in neuron-derived exosomes in PD patients and the disease progression. Thus, CNS-plasma exosomal tau could be a marker for PD diagnosis [37].

### 3.3. RNAs and Micro RNAs

LncRNAs, as transcribed RNA molecules, are involved in the regulation of gene expression through epigenetic, translation, transcription and post-transcription mechanisms [52,53]. On the other hand, micro RNAs (miRNAs) are small noncoding RNA molecules whose length is between 19 to 22 nucleotides and are relevant because of their regulatory role in gene expression through post-transcriptional processes [33]. Some examples of dysregulated miRNAs analyzed in this work were miR-1, miR-153, miR-409-3p, miR-19b-3p, miR-10a-5p and let-7g-3p. These were identified as useful screeners to discern PD subjects from healthy controls [20]. So is the case of pure SEVs miR-34a-5p levels that were higher in PD patients even at the beginning stage of PD when the disease duration was less than 5 years [26]. Parallel to that are miR-24, miR-195 and miR-19b quantified in serum exosomes, which may be useful noninvasive biomarkers for the diagnosis of PD [31]. Caggiu et al. found that miR-155-5p was upregulated, with a significant role in the inflammatory response to α-syn in the CNS [54]. Unlike miR-155-5p, miR-146a was downregulated in the PD group, affecting the regulation of the monocyte inflammatory response. miR-125b-5p was found to be downregulated, with a direct impact on the expression of the BDNF-AS molecule, capable of promoting autophagy and apoptosis in MPTP-induced PD [55]. It is thought that lncRNAs may influence the expression of target miRNAs, therefore promoting the development of PD [56].

In this review, LncRNAs MSTRG.242001.1 and MSTRG.169261.1 were highly expressed among PD patients, while Lnc-MKRN2-42:1 could be linked to the severity of PD symptoms such as dyskinesia and dysarthria and correlated with MDS-UPDRS III score [13].

LncRNAs and miRNAs have been demonstrated to have implications in the regulation of genes that have been linked to familial PD, which comprises less than 10% of the cases, as follows: (i) Mutations in the SNCA, LRRK2, PARK2 (Parkin), PARK 6 (PINK1) and PARK7 (DJ-1) genes are involved in physiological functions such as kinase signaling, ubiquitin-mediated protein degradation and mitochondrial respiratory chain function [57]. (ii) The LRRK2 autosomal dominant mutation has been established as the most common cause of familial PD. Higher levels of exosomal Ser(P)-1292 LRRK2 have been linked to PD and the presence of nonmotor symptoms [19].; (iii) R1441C, Y1699C and G20192 LRRK2 mutations increase autophosphorylated LRRK2 protein levels at the Ser-1292 residue [19]. (iv) The R1441C mutation appears to promote the formation of abnormally large MVBs which release more exosomes, thus increasing the presence of toxic forms of α-syn in the extracellular space that promote disease spread [9]. (v) Ravanidis et al. [58] reviewed the literature regarding the dysregulation of several miRNAs in relation with their target proteins and their role in PD, finding evidence of the alterations in miR-205 (LRRK2), miR-7 and mir-153 (SNCA), miR-22 (GBA), mir-544 (DJ-1), among others. (vi) PINK1 participates in different processes regarding mitochondria, such as quality control and damage regulation and has been described as a causative gene in the pathogenesis of PD [52,53]. (vii) Unlike the mutations mentioned above, the GBA heterozygous mutation was initially linked to PD through clinical observations in populations such as the Ashkenazi Jews [59]. GBA transcribes the GCase protein whose main function is the degradation of glucocerebroside into ceramide and glucose, as well as cleaving glucosyl sphingosine. The presence of the GBA1 mutation reduces the enzymatic activity of GCase, which triggers the unfolded protein response and is linked to endoplasmic-reticulum-associated degradation [60]. (viii) The combination of GCase activity, plasma L1CAM exosomal Linc-POU3F3 and plasma L1CAM exosomal α-syn has been shown to be more reliable (AUC 0.824) in distinguishing PD vs. controls than each individually [28].

The increasing study of RNAs related to PD elucidates their stability characteristics and although numerous factors are involved in their expression profile, they can be correctly quantified by achievable, reproducible methods, leading these to be potential biomarkers for early diagnosis and disease stage [61].

### 3.4. Neural-Derived Extracellular Vesicles

Neural-derived extracellular vesicles (NDEVs) and their cargo could be the most representative image of the actual state of a neurodegenerative disease such as PD, hence the constant search for specific markers.

EVs are secreted by all types of cells including neurons. NDEVs seem to reflect the brain status in neurodegenerative diseases and also potentially mediate the seeding of pathogenic forms of prone proteins such as α-syn in healthy neurons [62].

Numerous researchers have isolated exosomes containing L1CAM from plasma of PD patients to analyze their cargo in order to find a reliable biomarker with a noninvasive sample. From levels of α-syn in plasma neuronal exosomes, tau, LncRNA, GCase activity and DJ-1 are useful for the quantification of neuron-derived exosomes and oligodendrocyte-derived via ELISA, which could help its use as a biomarker for the diagnosis and progression of PD [18,28,32,37,39]. However, recently Norman et al. evaluated the use of L1CAM for the isolation of NDEV in plasma and CSF and advised against its use as a reliable marker due to its behavior as a soluble protein and not as a specific marker for EVs, explaining its nonspecific binding with soluble proteins such as α-syn [63]. In addition, the MISEV 2018 (Minimal Information for Studies of EVs 2018 guidelines) does not propose any biological marker to differentiate subtypes of EVs or to know their cell of origin [7].

### 3.5. Other Potential Biomarkers

The identification of specific proteins that are involved in PD pathogenesis has brought a new insight in the search for new biomarkers for early PD diagnosis and prognosis. Proteomic technology allows the study of the protein signature of EVs whether in normal conditions or in pathologic scenarios such as PD via different biofluids.

The expression of several proteins in EVs has been studied and analyzed. Jiang et al. found the C1q complex decreased PD patients’ serum exosomes. C1q is the recognition molecule key which contributes to the innate immune defense and regulates the adaptive immune response for the neuroprotection of the CNS and mediation of the formation of central synapses. Although linked to PD, it is still very difficult to elucidate the total role of the expression of C1q in the development of this disease [17,64].

As mentioned above, proteomics of EV derived from the CNS, erythrocytes or different biofluids have thrown hints for the study of new proteins that are involved in neurological mechanisms.

Lamontagne-Proulx et al. [22] performed a proteomic analysis of EVs derived from erythrocytes. In total, 8 out of the 818 proteins identified in the proteome of EVs had expressions that were significantly different in PD patients with various stages. Among these, QDPR is a key catalyzer for the recycling of BH4 (tetrahydrobiopterin), an essential cofactor in the biosynthesis of serotonin and precursors of L-dopa and 5-hydroxy-L-tryptophan (5-HTP) [65]. On the other hand, genetic variations in the USP24 protein coding gene, a member of the ubiquitin-specific protease family, is associated with the risk for late-onset PD [66].

Gualerzi et al. [23] studied the biochemistry of EVs through Raman spectroscopy and found there were biochemical differences between circulating EVs involving proteins, lipids and saccharides, which made it possible to discern between PD patients from HC with an accuracy of 71%. These findings could be useful to evaluate the effectiveness of rehabilitation and pharmacological treatments of PD in the future.

Blood-derived exosomal clusterin, complement C1r subcomponent, afamin, angiotensinogen variant, apolipoprotein D (ApoD), gelsolin and PEDF were progressively upregulated from mild to severe PD [17]. It is of major interest to analyze the molecular function of each protein and their interaction with upregulated pathways related to PD disease.

Apolipoprotein A1 (Apo A1), clusterin, complement C1r subcomponent and fibrinogen gamma chain exosomal expression levels in plasma of PD subjects may serve as a biomarker for the diagnosis of PD while Apo A1 could be of use in the future to measure the progression of the disease [27,35].

### 3.6. Neuroinflammation and Neurodegeneration

Neuroinflammation has been recognized as a key mediator in PD progression. There appears to be an alteration in the normal morphology of glial cells, including astrocytes and microglia, as well as an increase in inflammatory mediators in the parenchyma. This glial reaction is thought to happen due to neuronal death or dysfunction [67,68]. Exosomes containing pathologic forms of α-syn may activate microglia cells, promoting the accumulation and transmission of α-syn, while contributing to neuroinflammation by releasing inflammatory mediators [38].

The autophagy–lysosomal pathway (ALP) is capable of degrading aggregated misfolded proteins, such as α-syn, which in pathologic forms cause a neurodegenerative event when taken by microglia [69]. The dysfunction of ALP may come from an activation of microglia cells by exosomes containing pathologic forms of α-syn, leading to an accumulation and transmission of this protein into the system [38].

In addition to the mechanisms mentioned above, it has been hypothesized that the elevation of phosphorylated Ser-1292 in PD patients with advanced cognitive impairment is related to higher levels of plasma C-reactive protein levels as well as other inflammatory cytokines that eventually exacerbate neurodegeneration [5,14].

## 4. Conclusions

The interest in EVs has recently increased as it has been demonstrated that they play an important role in the multiple pathways that lead to neurodegenerative disorders such as PD. Their neuroprotective and neurotoxic functions have been meticulously studied. They intervene in the clearing of misfolded proteins by releasing detoxifying substances but also participate in the accumulation and transport of the origin cell cargo to specific cells, including misfolded proteins, nucleic acids and other cell constituents [8,12].

It is worth emphasizing that of all studies in this review, the largest total sample size was of 303 participants [37]. Future research projects should be performed in larger and independent cohorts with longitudinal approaches.

EVs isolation methods were varied, using centrifugation, differential centrifugation, ultracentrifugation, precipitation, or a combination of these, magnetic bead separation, size exclusion chromatography and fluorescence-activated cell sorting. Subsequent validation of the isolated exosomes in these studies was done using a variety of techniques such as Western blot, ELISA, TEM or NTA. The variation in the results found in this field clearly manifests the urgent need of more standardized protocols in EVs experimentation but also manifests the fantastic advances that have emerged with it.

Current data shed light on exosomal proteins, such as α-syn and DJ-1, as well as miRNAs and lncRNAs, as potential biomarkers that could be used to create standardized protocols for early PD diagnosis as well as tools to evaluate the severity and progression of the disease. Meaningful contributions such as the proteomic analysis of EVs and the study of its biochemistry as a fingerprint signature have marked a path for new biomarkers in this complex neurodegenerative disease. However, the future should be focused on finding reliable surface markers, using standardized procedures for the correct isolation of NDEVs from noninvasive biological samples such as saliva, which has demonstrated a good performance in early PD diagnosis, through the quantification of phospho-α-syn from neural-derived exosomes. Knowing the origin of EVs remains a challenge in research hence more studies are needed so we can resolve the puzzle correctly and understand every piece and pathway in PD.

In fact, miRNAs isolated from exosomal CSF (miR-1, miR-153, miR-409-3p, miR-19b-3p, miR-10a-5p and let-7g-3p) have shown great performance in distinguishing between PD patients and HC subjects. Even though it is an invasive procedure, lumbar puncture remains the most approachable way to study CNS. However, miR-24, miR-195 and miR-19b, all measured together in serum exosomes of PD patients, presented an AUC of 0.946 (95% CI, 0.910–0.981), the specificity was 90.0% and the sensitivity was 85.3%, so these may be useful noninvasive biomarkers for the diagnosis of PD. Future research is needed to identify the biological function of miR-24, miR-195 and miR-19b but for now, they shed light into this journey and create a new basis for new approaches.

The future should be focused on finding future biomarkers that not only allow us to make an early diagnosis of the disease, but also to distinguish PD from other neurodegenerative disorders.

## Figures and Tables

**Figure 1 ijms-23-11508-f001:**
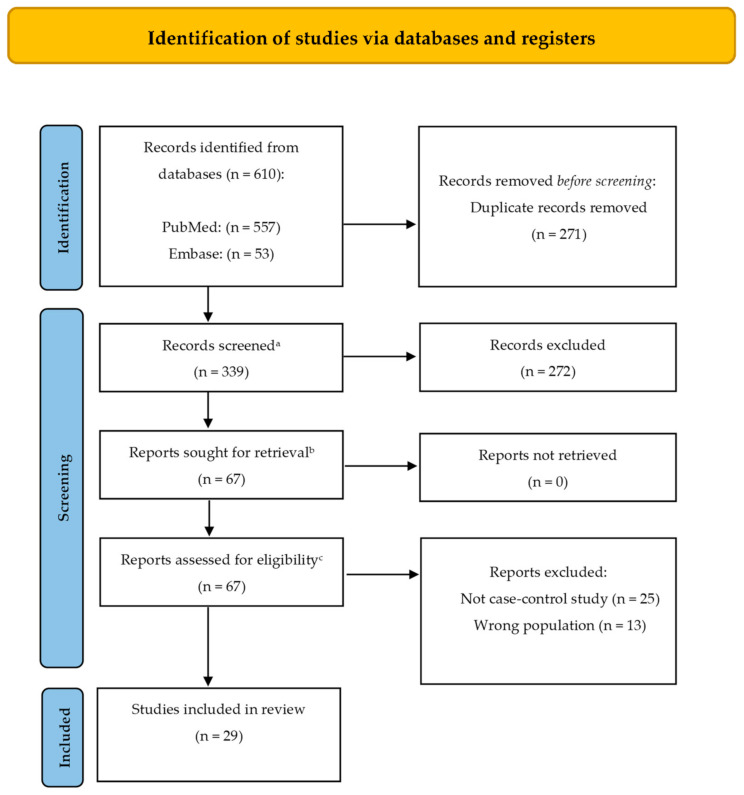
PRISMA (Preferred Reporting Items for Systematic Reviews and Meta-Analyses) 2020 flow diagram representing the searching and selection process for this review [15]. ^a^ This is the number of records identified minus the number from the duplicates removed box; ^b^ This is the number of articles obtained in preparation for full text screening (subtract the number of excluded records from the total number screened (previous step) to obtain the number sought for retrieval); ^c^ This should be the number of reports sought for retrieval minus the number of reports not retrieved (previous step).

**Figure 2 ijms-23-11508-f002:**
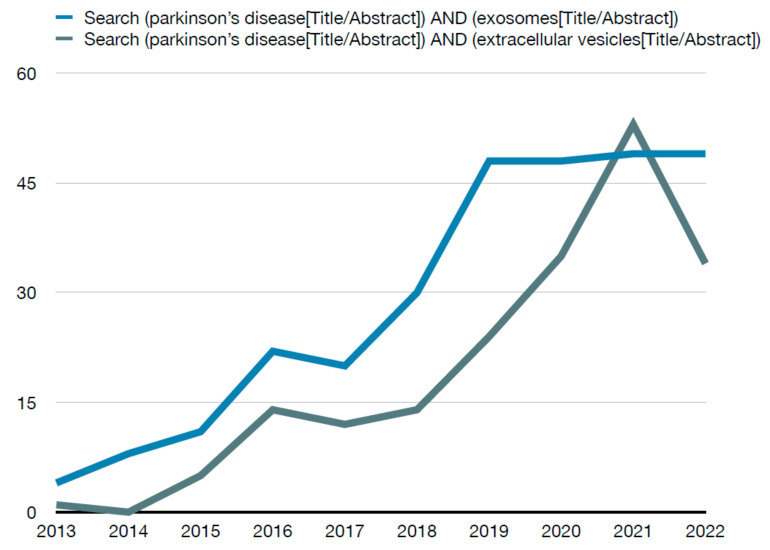
Number of PubMed results per year for the search method.

**Table 1 ijms-23-11508-t001:** Summary of studies analyzed and their main findings.

Object of Study	Sample Size * (Total/PD)	Type of Biological Sample	EVs Isolation Method	Main Outcomes	Ref.
Plasma exosomal prion proteins	60/40	Blood	Centrifugation and ExoQuick Plasma prep and exosome precipitation kit	Plasma exosomal prion protein levels were higher in PD group than in HC. It was also elevated in the PD-CI group compared to the PD-NCI group. Its concentration increased with age in HC, but no correlation was found in the PD group. It was positively associated with impaired cognitive level, visual spatial function, memory, attention and calculation abilities. Thus, plasma exosomal prion protein could be used as a biomarker for cognitive decline in PD patients.	[16]
Serum-derived exosome protein content	30/20	Blood	Ultracentrifugation	A total of 429 proteins were detected. Among these, 9 proteins were only detected in serum exosomes of patients with severe PD: protein S100, tyrosine protein kinase receptor, lactoferrin, dermcidin, platelet-activating factor acetyl hydrolase and isocitrate dehydrogenase. Clusterin, complement C1r subcomponent, afamin, angiotensinogen variant, apolipoprotein D, gelsolin, and PEDF were progressively upregulated from mild to severe PD. On the other hand, human neuroblastoma full-length cDNA clone CS0DD006YL02, precursor (AA-19 to 113), complement C1q subcomponent, myosin-reactive immunoglobulin kappa chain, Ig kappa chain V-III region, immunoglobulin mu chain, and immunoglobulin kappa variables 1 to 33 were gradually downregulated from mild to severe PD. A proteomic approach could be used to detect variations in the protein profile in different stages of PD development. Larger cohorts are needed.	[17]
DJ-1 and α-syn in plasma neural-derived exosomes	78/39	Blood	Centrifugation and precipitation	DJ-1 and α-syn levels from plasma neural-derived exosomes so as the ratio of plasma neural-derived exosomal DJ-1/total DJ-1 were significantly higher in the PD group than HC. No differences were observed between DJ-1 and α-syn levels from plasma neural-derived exosomes between PD patients at different stages of disease. They could serve as biomarkers for PD detection.	[18]
Exosomal Ser(P)-1292 LRRK2	158/79	Urine	Differential ultracentrifugation	Exosomal Ser(P)-1292 LRRK2 levels were significantly elevated in the PD group compared to HC; it was also positively correlated with multiple nonmotor measures of PD (MoCA, MDS-UPDRS Part I and II, Epworth SS).	[19]
Exosomal miRNA in CSF	102/47	Cerebrospinal fluid		Of a total of 746 exosomal miRNAs profiled, 27 of them were differentially expressed in CSF from PD patients compared to HC. Among them, 16 were upregulated (hsa-mir-103a, hsa-mir-30b, hsa-mir-16-2, hsa-mir-26a, hsa-mir-331-5p, hsa-mir-153, hsa-mir-132-5p, hsa-mir-485-5p, hsa-mir-127-3p, hsa-mir-409-3p, hsa-mir-433, hsa-mir-370, hsa-let-7g-3p, hsa-mir-873-3p, hsa-mir-136-3p, hsa-mir-10a-5p) and 11 were downregulated (hsa-mir-1, hsa-mir-22, hsa-mir-29, hsa-mir-374, hsa-mir-119a, hsa-mir-126, hsa-mir-151, hsa-mir-28, hsa-mir-301a, hsa-mir-19b-3p, hsa-mir-29c). Through DIANA-miRPath, the dysregulated exosomal miRNA signatures were associated with the neurotrophin signaling pathway, mTOR signaling pathway, ubiquitin-mediated proteolysis, long-term potentiation, axon guidance, cholinergic synapse, gap junction, dopaminergic synapse and glutamatergic synapse. Six miRNAs were selected for further validation (miR-1, miR-153, miR-409-3p, miR-19b-3p, miR-10a-5p, let-7g-3p) confirming that these miRNAs highly discriminated PD patients from HC.	[20]
Immune profiling of plasma-derived EVs	63/27	Blood	MACSPlex Human Exosome Kit	Plasma EV concentration was higher in patients with PD. Sixteen markers showed differences between the two PD and HC: CD4, CD19, CD45, CD1c, CD2, CD11c, CD31, CD41, CD42a, CD62, CD146, melanoma-associated chondroitin sulfate proteoglycan (MCSP), CD25, CD40, CD20 and HLA-ABC. Among the association of EV surface antigens as discriminants for PD diagnosis (eleven markers total), six were exclusive of the PD group (CD1c, CD11c, CD19, CD41b, CD45 and CD146). Through supervised machine learning algorithms, the combination of multiple EV specific markers showed a high sensitivity and specificity for the diagnosis of PD (AUC 0.908). Machine learning algorithms, based on EV-specific signature, discriminated patients with PD and MSA.	[21]
Profile and protein signature of EV in plasma and EV derived from erythrocytes	97/60 **	Blood	Centrifugation/fluorescence-activated cell sorting (FACS, Canto II Special Order Research Product)	A large cohort and patients in early stages of the disease are essential for future study. There was no difference in the number of EEV containing α-Syn and phospho α-Syn (serine 129) nor in the levels of α-Syn between PD patients and HC. Proteomic analyses of EEV of PD patients were performed. A total of 818 proteins were identified by removing hemoglobin from erythrocytes (in contrast to 356 without removing hemoglobin). Eight of these proteins were highly expressed and mapped the disease stage according to UPDRS scores, allowing them to be divided in three groups: Group I were highly expressed in controls (ABHD14B, alpha/beta hydrolase domain-containing protein 14B; AIDA, axin interactor dorsalization-associated protein; NADSYN1, glutamine-dependent NAD(+) synthetase) Group II expressed in mild PD patients (QDPR, quinoid dihydro pteridine reductase; AKR1A1, alcohol dehydrogenase NADP+; NRIP1, cannabinoid receptor-interacting protein 1. Group III predominantly expressed in moderate PD patients (USP24, ubiquitin carboxyl-terminal hydrolase 24; ATP5A1, ATP synthase subunit alpha mitochondrial).	[22]
Biochemistry of circulating EVs by Raman spectroscopy (RS)	40/22	Blood	Size exclusion chromatography/ultracentrifugation	Raman spectroscopy analysis demonstrated there were biochemical differences between circulating EVs from the PD group vs. HC, in particular, involving proteins, lipids and saccharides. RS could discern between PD patients from HC with an accuracy of 71% (*p* = 0.013). Blood-derived EVs from PD patients’ biochemical signature can be correlated to clinical scores measured by HY (describes degree of progression) and UPDRS part III (describes motor impairment) scores.	[23]
α-syn in salivary exosomes	134/74	Saliva	Centrifugation, precipitation and XYCQ Enrichment Kit	α-syn oligomer levels (2.05 pg/ng) and α-syn oligomer/α-syn total ratio (0.18 pg/ng) in salivary exosomes are higher in PD group than in HC group and may serve as a potential diagnostic biomarker for PD.	[24]
α-syn in plasma neuronal exosomes	94/53	Blood	Antibody-coated superparamagnetic microbeads	α-syn levels in plasma neuronal exosomes were significantly higher in patients with early stage PD compared with HCs (*p* = 0.007). Moreover, its concentrations had correlation with UPDRS III/(I,II,III) scores, NMSQ scores and SS-16 scores of patients with PD. α-syn levels in plasma neuronal exosomes could distinguish between early stage PD patients and HCs (AUC, 0.8; sensitivity, 100%; specificity, 57.1%). In a longitudinal study (n = 18 early stage PD), an increase in neuronal exosomal α-syn levels was associated with a higher risk of motor progression.	[25]
Circulating miR-34a-5p in small extracellular vesicles (SEVs)	29/15	Blood	Ultracentrifugation/density gradient centrifugation	miR-34a-5p levels were significantly overexpressed in pure SEVs from the plasma of PD patients compared to controls. In addition, miR-34a-5p expression in pure SEVs revealed a good ability to distinguish PD patients from control subjects (AUC, 0.738) suggesting its potential consideration as a marker of diagnosis at a molecular level. Pure SEVs miR-34a-5p levels were higher in PD patients even at the beginning stage of PD when the disease duration was less than 5 years. High levels of pure SEV miR-34a-5p were detected in PD patients with mild/progressive symptoms of disease and were associated with minimal/absent depression. There is the necessity to consider not only the whole plasma, but each EV subpopulation in order to improve the possibility to identify relevant differences of specific miRNAs levels. For this purpose, the SEVs purification protocol is crucial.	[26]
Brain-derived exosomes in plasma	52/15	Blood	Centrifugation	Plasma levels of brain-derived exosomes (BDE) were significantly higher in advanced PD compared to the HC group. Plasma levels of neuron-derived exosomes (NDE) and oligodendrocyte-derived exosomes (ODE) were higher even in mild PD compared to the HC group. These results suggested the capability of NDE and ODE as diagnostic biomarkers for PD. ODE levels were significantly higher in moderate to advanced disease, indicating it could be a biomarker for monitoring disease progression.	[27]
Linc-POU3F3 and α-syn levels in L1CAM exosomes and GCase activity.	178/93	Blood	Ultracentrifugation/antibody-coated superparamagnetic microbeads	The increase of L1CAM exosomal Linc-POU3F3 levels in plasma PD patients was positively correlated with disease severity (H-Y score and UPDRS-III). The L1CAM exosomal α-syn concentration and the ratio of exosomal α-syn to total α-syn (exo/total) were significantly increased in PD patients compared with healthy controls. GCase activity levels were decreased in PD vs. controls and negatively correlated with increased Linc-POU3F3 levels in the exosomal L1CAM of PD patients. The combination of plasma L1CAM exosomal Linc-POU3F3, plasma L1CAM exosomal α-syn and GCase activity showed to be more reliable (AUC 0.824) in distinguishing PD vs. controls than each individually. L1CAM exosomal Linc-POU3F3, α-syn levels in L1CAM exosomes and GCase activity may be potential diagnostic biomarkers and useful tools to evaluate the severity of PD.	[28]
Plasma EV α-syn	162/116	Blood	Size-exclusion chromatography (exoEasy Maxi kit)	Plasma EV α-syn levels were significantly decreased in the PD vs. control group and showed a negative association with akinetic rigidity syndromes severity in PD patients. Future large cohorts and investigations are necessary.	[29]
Exosomal α-syn from CSF	134/76	Cerebrospinal fluid	Centrifugation and ultracentrifugation	Total α-syn levels in CSF were significantly lower in the PD group than in the HC group. The CSF exosomal levels of α-syn were lower in the PD group than in the HC group. CSF exosomal levels of α-syn could serve as diagnostic biomarkers.	[30]
miRNAs in serum exosomes	149/109	Blood	Centrifugation/exosome isolation reagent from body fluids (Invitrogen)	A total of 24 previously reported miRNAs were analyzed in PD patients and an HC group. Among these, three had consistent results. The levels of miR-24 and miR-195 were significantly higher in serum samples from the PD group vs. HC (*p* < 0.05). On the contrary, miR-19b levels were significantly lower in PD vs. HC serum samples (*p* < 0.05). The sensitivity and specificity for diagnostic value in PD were: miR-195, 82.6% and 55.0%; miR-19b, 68.8% and 77.5; and miR-24, 81.7% and 85.0%, respectively. The three miRNAs (miR-24, miR-195, miR-19b) together, presented an AUC of 0.946 (95% CI, 0.910–0.981); the specificity was 90.0% and the sensitivity was 85.3%. The levels of miR-195, miR-19b and miR-24 may be useful noninvasive biomarkers for the diagnosis of PD. Future research is needed to identify the biological function of miR-24, miR-195 and miR-19b.	[31]
Neuronal exosomal α-syn	40/20	Blood	Immunoaffinity-based technology (magnetic beads coated with zwitterionic polymer pCBMA), conjugated with anti-L1CAM antibody.	The concentration of exosomes extracted by the precipitation method was significantly higher compared to the one obtained with ultracentrifugation. pCBMA@Fe3O4 MBs (magnetic beads coated with zwitterionic polymer pCBMA) conjugated with the anti-L1CAM antibody were effective in isolating neuronal-derived exosomes from serum and allowed them to perform electroanalyses with lower levels of serum compared to ELISA or electrochemiluminescence. The total quantification of α-syn using EIS was higher than the one detected by electrochemiluminescence. Levels of α-syn in neuronal exosomes in PD were higher compared to control. Neuronal exosome-associated Synt-1 content did not show any difference between the two groups.	[32]
Exosomal miRNAs	100/52	Blood	PureExo Exosome Isolation Kit (precipitation) and centrifugation	Exosomal miR-331-5p was significantly higher in PD patients than in the HC group. It was found mainly in exosomes. It is thought that miR-331-5p is transferred to PD-related cells through exosomes, being involved in the pathological process of PD.	[33]
Oligomeric α-syn, phosphorylated α-syn, and total α-syn in plasma exosomes	72/36	Blood	Precipitation (Total Exosome Isolation kit, Invitrogen) and differential centrifugation	The ratio of plasma exosomal α-syn/total α-syn monomers was significantly lower in PD patients vs. controls compared to higher levels in PD vs. controls in the ratio of α-syn/total α-syn oligomers and p-α-syn/p-α-syn oligomers. Triton X-100 insoluble α-syn and p-α-syn in the PD group was significantly higher than in HC compared to the soluble α-syn components. After treatment with different concentrations of PK, plasma exosomal insoluble α-syn components in the PD group were higher compared to controls. p-α-syn in PD patients’ plasma exosomes was more difficult to degrade by PK than that of the healthy controls. The levels of α-syn oligomer/total α-syn and p-α-syn oligomer/total p-α-syn in plasma exosomes of PD patients were higher than in the HC group. These results may indicate the PD pathological changes. ROC performance of both α-syn oligomer/total α-syn and p-α-syn oligomers/total p-α-syn (0.71, 0.69, respectively) in exosomes was moderate and may be a helpful tool in PD diagnosis.	[34]
Plasma-derived exosome protein content	24	Blood	Size exclusion chromatography on drip column (EV-Second)	Exosomal apolipoprotein A1 levels in PD patients at HY stage III were significantly decreased compared to HY stage II patients and correlated with the progression of the disease. Apolipoprotein A1, clusterin, complement C1r subcomponent and fibrinogen gamma chain exosomal expression levels may serve as a biomarker for disease progression.	[35]
Salivary exosomes from neuronal origin and its α-syn levels	36/18	Saliva	Centrifugation	Salivary exosomal phospho α-syn levels were higher in the PD group than in the HC group due to the higher secretion of exosomes from neuronal endings in salivary glands in the PD group. It could serve as a biomarker for early PD detection and a tool to measure disease progression in drug efficacy studies.	[36]
Central nervous system (CNS) exosomal tau in peripheral blood	303/91	Blood and cerebrospinal fluid	Centrifugation	Mean plasma exosomal tau was significantly higher in the PD group than in the HC group. In the PD group, plasma exosomal tau correlated with CSF t-tau and p-tau. Tau in L1CAM-containing exosomes was associated with disease duration. CNS-derived tau species could be used as PD biomarkers in plasma exosomes.	[37]
lncRNA in peripheral exosomes	14/7	Blood	Ultracentrifugation	A total of 15 upregulated and 24 downregulated lncRNAs were found. Of those, MSTRG.242001.1 and MSTRG.169261.1 were highly expressed among PD patients and MSTRG.336210.1 and lnc-MKRN2-42:1 among HC. A GO analysis of these lncRNAs showed their involvement in intracellular part, single-organism cellular process and heterocyclic compound binding. lnc-MKRN2-42:1 was found to regulate genes involved in apoptosis, synaptic remodeling, long-term potential, immunity and glutamate neurotransmitter metabolism. Thus, it was selected for further analysis, which showed that its expression was correlated with the MDS-UPDRS III score among PD patients, measuring the severity of dyskinesia and dysarthria.	[13]
Plasma exosomal α-syn	35/20	Blood	Differential centrifugation/ultracentrifugation	Exosomal total α-syn was elevated in plasma from the PD group compared to HC. Levels of exosomal α-syn oligomers and monomers in plasma were higher in the PD group than in HC group. The identification in plasma exosomal α-syn oligomers could possibly be used as a potential biomarker.	[38]
L1CAM-exosomal α-syn from CNS	77/38	Blood	Centrifugation and ExoQuick exosome precipitation solution	The mean value of α-syn in L1CAM-containing exosomes was lower in the PD group than in the ET and HC groups. It was lower in NTD-PD (nontremor-dominant group) compared to the TD-PD group (tremor-dominant group). CNS-derived exosomal α-syn in serum may be inverse to the course and severity in PD patients, helping diagnose PD patients and differentiating motor types in early stages.	[39]
Exosomal OxiDJ-1 in urine	55/33	Urine	Centrifugation and filtration	OxiDJ-1 levels in urine were higher in the PD group than HC. It could serve as a diagnostic biomarker for PD diagnosis.	[40]
Plasma exosomal α-syn	72/39	Blood	Centrifugation and ultracentrifugation	α-syn concentration in plasma exosomes was higher in PD patients than HC, confirming it is associated with the pathological status.	[14]
Poly (ADP-Ribose) and α–syn extracellular vesicles	117/57	Blood	Centrifugation/incubation with fluorescent-labelled primary antibodies against total α-syn	Median concentration of α-syn extracellular vesicles was significantly higher in PD patients compared to the other groups (Kruskal–Wallis, *p* < 0.0001).	[41]
α-syn conformers	80/30	Blood	Centrifugation and precipitation	The detection of pathological α-syn conformers from neuron-derived extracellular vesicles from blood plasma samples has the potential to evolve into a blood-biomarker of PD	[42]

* Sample size refers to the number of all participants whose biological samples were taken and used for investigation purposes. EVs: extracellular vesicles. ** Comparison cohort of 107 Huntington’s disease patients plus HC and an independent cohort of 42 PD patients.

**Table 2 ijms-23-11508-t002:** Summary of potential biomarkers in the analyzed studies.

Potential Biomarkers	Specifications	Type of Biological Sample	Ref.
α-syn	Plasma neural-derived exosomes	Blood	[18]
α-syn	Plasma neuronal exosomes	Blood	[25]
α-syn	Plasma EVs	Blood	[29]
α-syn	CSF exosomal levels	Cerebrospinal fluid	[30]
α-syn	Neuronal exosomes	Blood	[32]
α-syn conformers	Plasma EVs	Blood	[42]
Oligomeric α-syn and total α-syn	Salivary EVs	Saliva	[24]
Oligomeric α-syn, phosphorylated α-syn and total α-syn	Plasma exosomes	Blood	[34]
Phosphorylated α-syn	Neuronal exosomes	Saliva	[35]
Poly (ADP-Ribose) and α–syn	Plasma EVs	Blood	[41]
PrPc (cell prion protein)	Plasma exosomal prion protein	Blood	[16]
DJ-1	Plasma neural-derived exosome	Blood	[18]
OxiDJ-1	Urine exosomes	Urine	[39]
tau protein	L1CAM exosomal tau in plasma and cerebrospinal fluid	Blood and cerebrospinal fluid	[36]
miR-1, miR-153, miR-409-3p, miR-19b-3p, miR-10a-5p and let-7g-3p	miRNAs in cerebrospinal fluid (CSF) exosomes	Cerebrospinal fluid	[20]
miR-24, miR-195 and miR-19b	miRNAs in serum exosomes	Blood	[31]
miR 34a 5p	Circulating miR-34a-5p in small extracellular vesicles (SEVs)	Blood	[26]
miR-331-5p and miR-505	Plasma exosomes circulating miRNAs	Blood	[33]
lnc-MKRN2-42:1	lncRNA in plasma exosomes	Blood and cerebrospinal fluid	[13]
Ser(P)-1292 LRRK2	Autophosphorylated LRRK2 protein in urinary exosomes	Urine	[19]
Linc-POU3F3 and α-syn levels and GCase activity.	Plasma L1CAM exosomal levels	Blood	[28]
CD1c, CD11c, CD19, CD41b, CD45 and CD146	EVs surface antigens expression in plasma	Blood	[21]
NDEs (neuron-derived exosomes) and ODE (oligodendrocyte-derived exosomes)	Quantification of brain derived exosomes in plasma	Blood	[27]
Apolipoprotein A1, clusterin, complement C1r subcomponent and fibrinogen gamma chain	Plasma-derived protein content	Blood	[35]
Clusterin, complement C1r subcomponent, afamin, angiotensinogen variant, apolipoprotein D, gelsolin, PEDF, human neuroblastoma full-length cDNA clone CS0DD006YL02, precursor (AA-19 to 113), complement C1q subcomponent, myosin-reactive immunoglobulin kappa chain, Ig kappa chain V-III region, immunoglobulin mu chain and immuno-globulin kappa variables 1 to 33.	Proteomic analysis of serum exosomes	Blood	[17]
QDPR, Quinoid Dihydro pteridine Reductase; AKR1A1, alcohol dehydrogenase NADP+; NRIP1, cannabinoid receptor-interacting protein 1; USP24, ubiquitin carboxyl-terminal hydrolase 24 and ATP5A1, ATP synthase subunit alpha mitochondrial	Proteomic analysis of EVs derived from erythrocytes	Blood	[22]

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
