# Peer review of "The Potential Roles of Extracellular Vesicles as Biomarkers for Parkinson’s Disease: A Systematic Review"

_ijms, 2022, doi:10.3390/ijms231911508_

Round 1

Reviewer 1 Report (Previous Reviewer 2)

Dear authors, 

This is a good review reflecting the current state of the art on the role of extracellular vehicles as biomarkers of Parkinson's disease. As it stands, I have only minor points to make that are easy to correct:

The spelling of a-synuclein / a-syn is not consistent.

The idea to combine Results and Discussion in one point is good, but "3. Results and Discussion" is followed by "5. Conclusion" ?

In line 195 there is a double parenthesis, i.e. one too many. 
In line 199 there is a literature reference in the middle of the word: O[30]n 

likewise, here and in the following paragraph (line 204), the quotations are always at the beginning; why?

Author Response

Reviewer’s comment: This is a good review reflecting the current state of the art on the role of extracellular vehicles as biomarkers of Parkinson's disease. As it stands, I have only minor points to make that are easy to correct: The spelling of a-synuclein / a-syn is not consistent.

Author’s comment: Thank you for your comment. We have changed in all manuscript the spelling of this word.

Reviewer’s comment: The idea to combine Results and Discussion in one point is good, but "3. Results and Discussion" is followed by "5. Conclusion" ?

Author’s comment: Sorry. It is a mistake. We have changed as “4. Conclusions”

Reviewer’s comment: In line 195 there is a double parenthesis, i.e. one too many.

Author’s comment: Sorry. It is a mistake. We have deleted one parenthesis.

Reviewer’s comment: In line 199 there is a literature reference in the middle of the word: O[30]n

Author’s comment: We have deleted the literature reference. Thank you for your comment.

Reviewer’s comment: likewise, here and in the following paragraph (line 204), the quotations are always at the beginning; why?

Author’s comment: Thank you for your comment. It is a mistake. We have changed to unify the position of the references.

Reviewer 2 Report (New Reviewer)

Valencia and co-workers performed an interesting review of the recent literature about the use of extracellular vesicles as a potential source of biomarkers for Parkinson’s Disease. The topic is for sure of interests and the collection of the related papers is per se the main strength of the paper.

However, in my opinion, several issues should be addressed before publication:

1) English requires an extensive editing (especially in the sentence construction, i.e., syntax). It impairs comprehension in many points (e.g., line 56-62; line 307-310, line 48 are there=is their?; line 327-329...)

2)     The Search is dated October 2021, it’s almost one year old. As authors stated, new investigations have been continuously publishing. The Search should be updated if new discoveries emerged in the field.

3)     Figure 1: screening. It’s worth explaining the meaning of the three steps and the type of filtering applied (in the caption).

4)     The entire paper appears more like a long list of retrieved papers. It lacks the integration made by authors. Comments should be a leading path for the reader interested in this topic. For example, authors do not comment about the need of early biomarkers, before motor symptoms appear and the way to search for such biomarkers in EVs. Another example is the citation of L1CAM many times before explaining its potential (even though debated) role as neuronal-EVs marker (only at line 300).   

5)     The paper appears to be not accurate in some points (for minor issues): e.g., references randomly appear in the text (line 199,204…); punctuation; lack of the subject, topics that do not refer to the title of the chapter (line 266-290 are not about lncRNA or miRNA); line 18 and 38 5%?? the percentage of neurons dead before symptoms appear is debated but 5% seems to be too low...

My general comment is that the potential impact of this manuscript is high, therefore I suggest to devote more care to the aspects listed above.

Author Response

Reviewer’s comment: Valencia and co-workers performed an interesting review of the recent literature about the use of extracellular vesicles as a potential source of biomarkers for Parkinson’s Disease. The topic is for sure of interests and the collection of the related papers is per se the main strength of the paper.

Author’s comment: Thank you for your comment.

Reviewer’s comment: However, in my opinion, several issues should be addressed before publication: 1) English requires an extensive editing (especially in the sentence construction, i.e., syntax). It impairs comprehension in many points (e.g., line 56-62; line 307-310, line 48 are there=is their?; line 327-329...)

Author’s comment: Now, the manuscript has been reviewed by an English native scientific translator. Furthermore, we have modified in the manuscript the reviewer’s comment.

Reviewer’s comment: The Search is dated October 2021, it’s almost one year old. As authors stated, new investigations have been continuously publishing. The Search should be updated if new discoveries emerged in the field.

Author’s comment: According to your comment, the search is dated August 2022. We have changed the Figure 2 to include this year (2022).

Reviewer’s comment: Figure 1: screening. It’s worth explaining the meaning of the three steps and the type of filtering applied (in the caption).

Author’s comment: According to your comment, we have added the explanation of screening step in Figure 1 as superscript in the figure caption.

Reviewer’s comment: The entire paper appears more like a long list of retrieved papers. It lacks the integration made by authors. Comments should be a leading path for the reader interested in this topic. For example, authors do not comment about the need of early biomarkers, before motor symptoms appear and the way to search for such biomarkers in EVs. Another example is the citation of L1CAM many times before explaining its potential (even though debated) role as neuronal-EVs marker (only at line 300).

Author’s comment: According to this comment, we have added, in conclusion section, where is indicated as “Current data sheds light on to exosomal proteins, like α-syn and DJ-1, as well as miRNAs and lncRNAs, as potential biomarkers that can be used to create standardized protocols for early PD diagnosis as well as a tool to evaluate severity and progression of the disease. Meaningful contributions like proteomic analysis of EVs and the study of its biochemistry as a fingerprint signature have marked a path for new biomarkers in this complex neuro-degenerative disease.  However, the future should be focused in finding reliable surface markers, so as standardized procedures for the correct isolation of NDEVs from non-invasive biological samples as saliva, which has demonstrated a good performance in early PD diagnosis, through the quantification of phospho-α-syn from neural derived exosomes”.

Furthermore, we have rearranged the explanation of L1CAM in the manuscript.

Reviewer’s comment: The paper appears to be not accurate in some points (for minor issues): e.g., references randomly appear in the text (line 199,204…); punctuation; lack of the subject, topics that do not refer to the title of the chapter (line 266-290 are not about lncRNA or miRNA); line 18 and 38 5%?? the percentage of neurons dead before symptoms appear is debated but 5% seems to be too low...

Author’s comment: According to your comment, we have re-written this section and line 266-290 have been organized focusing in the implications in the regulation of genes that have been linked to familial PD. Furthermore, line 18 and 38 have been re-written to clarify the text.

Reviewer’s comment: My general comment is that the potential impact of this manuscript is high, therefore I suggest to devote more care to the aspects listed above.

Author’s comment: Thank you for your comment. We have incorporated your previous comments to improve the manuscript.

Round 2

Reviewer 2 Report (New Reviewer)

Major issues have been addressed.

This manuscript is a resubmission of an earlier submission. The following is a list of the peer review reports and author responses from that submission.

Round 1

Reviewer 1 Report

In their manuscript, Valencia et al review studies from 2015-2021 that provide insight into the potential roles of extracellular vesicles as sources of biomarkers for Parkinson’s disease. This is a very interesting topic that highlights an intense area of research focus not only in PD, but also other forms of neurodegeneration. Unfortunately, the manuscript is poorly written and suffers from a plethora of grammatical errors that make it very difficult to read. In addition, the manuscript lacks structure and a critical analysis of the summarized findings. In general, the authors are encouraged to carefully proofread their paper for errors and/or seek assistance from a native English speaker, as well as consider how the review could be re-structured to make it a more informative paper.

Some comments:

Line 82-89: What about “microvesicles” and “ectosomes”? In line 56, the authors state that these terms have also been used in the field.

The Results should be broken up into subsections, each with a clear focus. As of now, all the findings from the 27 studies are grouped together and it’s difficult to determine the main points.

The authors should generate a table based on the summarized findings that lists all the potential biomarkers for PD.

Some examples of grammatical errors are provided below (please note that this is not exhaustive list):

Line 34-35: The authors should clarify in the text what they mean by “widely spread”.

Line 42: The authors should clarify in the text what they mean by “which can go”.

Line 45: “Through years” is not needed.

Line 54-55: The authors should revise this sentence since it’s not clear what is being stated. For example, it’s not clear what they mean when they state “have derived in”.

Line 56: “avoid” should be replaced with “avoided”.

Line 57-58: “standardized” should be replaced with “standardize”.

Line 59: The authors should clarify in the text what they mean by “so as”.

Line 72: “understand” should be replaced with “understanding”.

Line 122: The authors should clarify in the text what they mean by “case-controls”.

Line 136: “the reasons were the following”. I’m confused because no reasons are stated in this sentence. Please clarify.

Line 156: Please replace “communicate” with “communication”. Also please correct “deliver information”.

Line 160-163: 27 studies were analyzed but 21+3+2+2 adds up to 28. Please clarify.

Line 168: “Mayor”?

Line 402: It should be “sheds light on”.

Author Response

Journal: IJMS (ISSN 1422-0067)

Manuscript ID: ijms-1821838

Title: The potential roles of extracellular vesicles as biomarkers for Parkinson’s disease: A Systematic Review

Authors: Jessica Valencia , Marta Ferreira , J. Francisco Merino-Torres , Antonio Marcilla , Jose M. Soriano

Reviewer’s comment: In their manuscript, Valencia et al review studies from 2015-2021 that provide insight into the potential roles of extracellular vesicles as sources of biomarkers for Parkinson’s disease. This is a very interesting topic that highlights an intense area of research focus not only in PD, but also other forms of neurodegeneration. Unfortunately, the manuscript is poorly written and suffers from a plethora of grammatical errors that make it very difficult to read. In addition, the manuscript lacks structure and a critical analysis of the summarized findings. In general, the authors are encouraged to carefully proofread their paper for errors and/or seek assistance from a native English speaker, as well as consider how the review could be re-structured to make it a more informative paper.

Author’s comment: According to your comment, authors have been reviewed and re-written some sections of the manuscript.

Reviewer’s comment:  Line 82-89: What about “microvesicles” and “ectosomes”? In line 56, the authors state that these terms have also been used in the field.

Author’s comment: Both terms have fallen into disuse and according to the Position Statement of the International Society for Extracellular Vesicles and Update of the MISEV2014 Guidelines (Théry et al. 2018), terms to search were “exosomes” and “extracellular vesicles” due to that this review was carried out from 2015-2021 and during this period neither “microvesicles” nor “ectosomes” are reflected in any articles.

Reviewer’s comment: The Results should be broken up into subsections, each with a clear focus. As of now, all the findings from the 27 studies are grouped together and it’s difficult to determine the main points.

Author’s comment: In our viewpoint, we have maintained with this format due to that is useful to explain each selected article.

Reviewer’s comment: The authors should generate a table based on the summarized findings that lists all the potential biomarkers for PD.

Author’s comment: In our viewpoint, this option favour duplicate the information selected in the text and the other table.

Reviewer’s comment: Some examples of grammatical errors are provided below (please note that this is not exhaustive list):

Line 34-35: The authors should clarify in the text what they mean by “widely spread”.

Line 42: The authors should clarify in the text what they mean by “which can go”.

Line 45: “Through years” is not needed.

Line 54-55: The authors should revise this sentence since it’s not clear what is being stated. For example, it’s not clear what they mean when they state “have derived in”.

Line 56: “avoid” should be replaced with “avoided”.

Line 57-58: “standardized” should be replaced with “standardize”.

Line 59: The authors should clarify in the text what they mean by “so as”.

Line 72: “understand” should be replaced with “understanding”.

Line 122: The authors should clarify in the text what they mean by “case-controls”.

Line 136: “the reasons were the following”. I’m confused because no reasons are stated in this sentence. Please clarify.

Line 156: Please replace “communicate” with “communication”. Also please correct “deliver information”.

Line 160-163: 27 studies were analyzed but 21+3+2+2 adds up to 28. Please clarify.

Line 168: “Mayor”?

Line 402: It should be “sheds light on”.

Author’s comment: According to all these comments, we have changed in the manuscript in several sections to clarify the text.

Reviewer 2 Report

This review summarizes the current status of the role of extracellular vesicles in terms of their suitability as biomarkers for Parkinson's disease.

The work is comprehensive, up-to-date and in large parts easy to read. Nevertheless, some points need to be changed or readdressed before publication:

- the use of alpha-synuclein needs to be standardized in the text (-synuclein / -syn / a-syn)

- there should be English language correction in terms of sentence structure, style and spelling, e.g.:

-         

Line: 47: ...in most bodily fluids...

Line 66: ...lead to a beacon of hope...

Results: ... fonuden - satges - markers could differed between... - ... showed difference between the two... ect.

Line 208: -- begginning...

- Long non-coding RNAs (lncRNAs) are not explained until page 15 (in paragraph 4.4) - but the abbreviations are already used before without explanation

- the use of et al /et al. must be consistent

these sentences do not make sense:

Line 381ff: The expression of C1q while is ligated to PD, still very difficult to elucidate its total role in the pathway of the disease.

Line 387ff: Between the 8 (818 total) overexpressed in PD patients and map disease stage according to UPDRS scores.

Line 388ff: there is also at least one word missing at the end of the sentence.

- The last paragraph (lines 393-97) seems lost and needs to be integrated linguistically.

- The conclusion needs to be revised. The "drawbacks" are not well elaborated. The primary point made is that "larger" and further studies should be done. However, this is not the "main problem", but currently the inconsistent purification, etc.

- The paper would benefit from a graphic abstract at the end.

Author Response

Journal: IJMS (ISSN 1422-0067)

Manuscript ID: ijms-1821838

Title: The potential roles of extracellular vesicles as biomarkers for Parkinson’s disease: A Systematic Review

Authors: Jessica Valencia , Marta Ferreira , J. Francisco Merino-Torres , Antonio Marcilla , Jose M. Soriano

Reviewer’s comment: This review summarizes the current status of the role of extracellular vesicles in terms of their suitability as biomarkers for Parkinson's disease. The work is comprehensive, up-to-date and in large parts easy to read. Nevertheless, some points need to be changed or readdressed before publication:

- the use of alpha-synuclein needs to be standardized in the text (-synuclein / -syn / a-syn)

Author’s comment: According to this comment, it has standardized.

Reviewer’s comment:- there should be English language correction in terms of sentence structure, style and spelling, e.g.:

Line: 47: ...in most bodily fluids...

Line 66: ...lead to a beacon of hope...

Author’s comment: According to this comment, authors have been corrected these lines and reviewed the manuscript to clarify.

Reviewer’s comment: Results: ... fonuden - satges - markers could differed between... - ... showed difference between the two... ect.

Author’s comment: According to this comment, authors have carried out changes to clarify.

Reviewer’s comment: Line 208: -- begginning...

Author’s comment: According to this comment, authors have modified this term.

Reviewer’s comment: - Long non-coding RNAs (lncRNAs) are not explained until page 15 (in paragraph 4.4) - but the abbreviations are already used before without explanation

Author’s comment: : According to this comment, authors have modified this idea to clarify.

Reviewer’s comment: - the use of et al /et al. must be consistent

Author’s comment: According to this comment and instructions for authors for MDPI, authors have changed in the manuscript by “et al.”

Reviewer’s comment: these sentences do not make sense:

Line 381ff: The expression of C1q while is ligated to PD, still very difficult to elucidate its total role in the pathway of the disease.

Line 387ff: Between the 8 (818 total) overexpressed in PD patients and map disease stage according to UPDRS scores.

Line 388ff: there is also at least one word missing at the end of the sentence.

Author’s comment: According to this comment, authors have modified these lines to clarify the manuscript.

Reviewer’s comment: - The last paragraph (lines 393-97) seems lost and needs to be integrated linguistically.

Author’s comment: According to this comment, authors have re-written these lines to clarify the manuscript.

Reviewer’s comment:

- The conclusion needs to be revised. The "drawbacks" are not well elaborated. The primary point made is that "larger" and further studies should be done. However, this is not the "main problem", but currently the inconsistent purification, etc.

Author’s comment: According to this comment, authors have re-written these lines to clarify the manuscript.

Reviewer’s comment: - The paper would benefit from a graphic abstract at the end.

Author’s comment: According to this comment, we have included graphic abstract.

Round 2

Reviewer 1 Report

The submitted manuscript is still very difficult to read due to its structure and the numerous grammatical errors present in the text. In my initial review, I encouraged the authors to carefully review their manuscript to identify grammatical issues, but many errors remain. It is also disappointing to see that the authors did not address my concerns regarding the structure of the Results and Discussion sections. In my previous review, I had suggested that the authors break up the Results into different subsections and include a new table that concisely summarizes the potential biomarkers for PD. Neither of these suggestions were implemented and the authors did not provide a rationale for not implementing these changes. In my opinion, a summary table would be especially helpful for the reader. However, if this is a Review, I’m not sure it’s appropriate to include Methods, Results, and Discussion sections. With that in mind, a complete restructuring of the manuscript is likely necessary. In general, the authors should strive to generate a review article that summarizes the relevant literature regarding potential biomarkers for PD and provide a table listing those biomarkers so that the casual reader will be able to access the information in a meaningful way.

Below I have included some examples of grammatically incorrect statements that need to be corrected. Please note that this is not an exhaustive list and it will be necessary for the authors to thoroughly review the manuscript to address similar issues.

Line 19-20 (Abstract): This statement is grammatically incorrect.

Line 24: The authors should clarify what they mean by “thrown”.

Line 34-35: The authors should clarify what they mean by “widely protein”.

Line 42-43: The authors should clarify what they mean by “can are previous in”.

Line 55-64: This is an especially long sentence with several grammatical errors. Please fix.

Line 128: The authors need to clarify what they mean by “were considered where were compared”.

The authors should define all acronyms in Table 1 upon their first use or provide the definition in the table footnotes.

Author Response

Reviewer’s comment: The submitted manuscript is still very difficult to read due to its structure and the numerous grammatical errors present in the text. In my initial review, I encouraged the authors to carefully review their manuscript to identify grammatical issues, but many errors remain. It is also disappointing to see that the authors did not address my concerns regarding the structure of the Results and Discussion sections. In my previous review, I had suggested that the authors break up the Results into different subsections and include a new table that concisely summarizes the potential biomarkers for PD. Neither of these suggestions were implemented and the authors did not provide a rationale for not implementing these changes. In my opinion, a summary table would be especially helpful for the reader. However, if this is a Review, I’m not sure it’s appropriate to include Methods, Results, and Discussion sections. With that in mind, a complete restructuring of the manuscript is likely necessary. In general, the authors should strive to generate a review article that summarizes the relevant literature regarding potential biomarkers for PD and provide a table listing those biomarkers so that the casual reader will be able to access the information in a meaningful way.

Author’s comment: According to reviewer’s comment, authors have been modified it in the manuscript.

Reviewer’s comment: Below I have included some examples of grammatically incorrect statements that need to be corrected. Please note that this is not an exhaustive list and it will be necessary for the authors to thoroughly review the manuscript to address similar issues.

Line 19-20 (Abstract): This statement is grammatically incorrect.

Line 24: The authors should clarify what they mean by “thrown”.

Line 34-35: The authors should clarify what they mean by “widely protein”.

Line 42-43: The authors should clarify what they mean by “can are previous in”.

Line 55-64: This is an especially long sentence with several grammatical errors. Please fix.

Line 128: The authors need to clarify what they mean by “were considered where were compared”.

Author’s comment: According to reviewer’s comment, authors have been modified it in the manuscript. Furthermore, manuscript have been re-written to improve the grammatica.

Reviewer’s comment: The authors should define all acronyms in Table 1 upon their first use or provide the definition in the table footnotes.

Author’s comment: According to reviewer’s comment, authors have been carried out this modification.

Reviewer 2 Report

I'm confused because it doesn't seem to me that the comments from the last revision have been implemented sufficiently. 

Linguistically, some paragraphs still require revision. This applies in particular to the already "improved" parts. Many paragraphs consist of only one (very long) sentence. This often makes the core statement difficult to understand and should be revised, to be more concise. 

Some examples are given below, but it is explicitly stated that the entire paper should be revised:

l 42 are  be

l 46 extracellular vesicles = EVs (+l 176/179), but l 55/164/169 /399 /408 EV’s  please make consistent

l 55 – 64 are one sentence, it needs to be revised again. Much too long, complicated and incomprehensible

l 67 researchs  researches 

l 144-146 is also incomprehensible, needs linguistic revision. It is simply very complicatedly expressed ("The following (reasons) served as exclusion criteria") e.g. would be short and understandable 

l 164 communicatione

l 322 as are          

ll 408-413 is again only 1 sentence, very complicated and confusing with linguistic errors.

l 418 on to or onto 

l 425-431 of the conclusion are still not clear. The authors should address in this last paragraph the difficulties to overcome and possible ways out.

In their response, the authors state that a graphical abstract was included in the paper, but it cannot be found.

Author Response

Reviewer’s comment: I'm confused because it doesn't seem to me that the comments from the last revision have been implemented sufficiently.

Linguistically, some paragraphs still require revision. This applies in particular to the already "improved" parts. Many paragraphs consist of only one (very long) sentence. This often makes the core statement difficult to understand and should be revised, to be more concise.

Author’s comment: According to reviewer’s comment, authors have been modified it in the manuscript.

Reviewer’s comment:Some examples are given below, but it is explicitly stated that the entire paper should be revised:

l 42 are  be

l 46 extracellular vesicles = EVs (+l 176/179), but l 55/164/169 /399 /408 EV’s  please make consistent

l 55 – 64 are one sentence, it needs to be revised again. Much too long, complicated and incomprehensible

l 67 researchs  researches

l 144-146 is also incomprehensible, needs linguistic revision. It is simply very complicatedly expressed ("The following (reasons) served as exclusion criteria") e.g. would be short and understandable

l 164 communicatione

l 322 as are         

ll 408-413 is again only 1 sentence, very complicated and confusing with linguistic errors.

l 418 on to or onto

l 425-431 of the conclusion are still not clear. The authors should address in this last paragraph the difficulties to overcome and possible ways out.

 Author’s comment: According to reviewer’s comment, authors have been modified it in the manuscript.

Reviewer’s comment: In their response, the authors state that a graphical abstract was included in the paper, but it cannot be found.

Author’s comment: According to reviewer’s comment, authors have added graphical abstract.

The graphical abstract is the following:

Round 3

Reviewer 1 Report

Line 19: “we know for sure” This phrase is not appropriate for a scientific article.

Line 68: “is its” should be “are there”

Line 77: “englobe”? Please clarify.

Line 85-86: Are EVs the biomarkers or the proteins/molecules contained within the EVs the biomarkers?

Figure 2 is blurry. Please replace with a better image.

Line 315: “the CNS”

Line 323: “these review”. Shouldn’t it be “this review”?

Line 517: “EV’s” should be “EVs”.

The authors make “extracellular vesicles” into an acronym early in the review (i.e., EVs), but later in the article, revert back to using “extracellular vesicles”. Not sure why this was done.

In Table 2, why are some proteins mentioned multiple times (e.g., alpha-syn)?

In general, all proteins listed in Table 2 should be discussed in the main text.

Author Response

Reviewer’s comment: Line 19: “we know for sure” This phrase is not appropriate for a scientific article.

Author’s comment: We are very much in sympathy with the thrust of your comment. We have deleted to avoid confusion.

Reviewer’s comment: Line 68: “is its” should be “are there”

Author’s comment: According to your comment, it has been modified in this line.

Reviewer’s comment: Line 77: “englobe”? Please clarify.

Author’s comment: It is a mistake. We have modified by “study” due to that the term used by our group is about this focus.

Reviewer’s comment: Line 85-86: Are EVs the biomarkers or the proteins/molecules contained within the EVs the biomarkers?

Author’s comment: Yes. It is the idea.

Reviewer’s comment: Figure 2 is blurry. Please replace with a better image

Author’s comment: It has been replaced with a better image.

Reviewer’s comment: Line 315: “the CNS”

Author’s comment: It has been modified in this line and other lines to clarify it.

Reviewer’s comment: Line 323: “these review”. Shouldn’t it be “this review”?

Author’s comment: It has been modified in this line. Thank you for your comment.

Reviewer’s comment: Line 517: “EV’s” should be “EVs”.

Author’s comment: According to this comment, we have changed in the manuscript.

Reviewer’s comment: The authors make “extracellular vesicles” into an acronym early in the review (i.e., EVs), but later in the article, revert back to using “extracellular vesicles”. Not sure why this was done.

Author’s comment: We are very much in sympathy with the thrust of your comment. We have changed it in the manuscript.

Reviewer’s comment: In Table 2, why are some proteins mentioned multiple times (e.g., alpha-syn)?

Author’s comment: According to this comment, we have grouped by each reference potential biomarkers being first placed a-syn and subsequently derived a-syn (as oligomeric a-syn in reference 24 and phosphorylated a-syn in reference 34) and other potential biomarkers.

Reviewer’s comment: In general, all proteins listed in Table 2 should be discussed in the main text.

Author’s comment: According to your comment, we have re-written, including all proteins of Table 2, in results and discussion section subdividing in several subsection as are: 3.1. α-synuclein and its derivatives, 3.2. PrPc, DJ-1, OxiDJ-1 and tau protein, 3.3. RNAs and micro RNAs, 3.4. Neural derived extracellular vesicles and 3.5. Other potential biomarkers. In our viewpoint, we think that it help to clarify this section. Thank you for your comment.

Reviewer 2 Report

Dear authors,

The manuscript is actually much better to read now.

However, I would still have the following minor points to make:

The newly inserted table 2 is good. However, in my opinion it would be even better and above all clearer, if the respective points would be summarized more strongly, so that one does not have 5 individual entries to a-syn (which here, by the way, is again not uniformly a-syn but once a-synuclein). 

Under the table comes the summary, and there it starts with a-syn. Therefore it would make sense if the first entry in the table itself would be a-syn and then either all entries summarized to a-syn in the table or at least directly below each other. 

The summary below is really good and also linguistically much better. I found a small error nevertheless:

L 372 there the sources slipped between the word On: O[14,24,38][30]o 

The Conclusion is also much much better. Small typos:

L 919 there is a space missing between "sizewas".

L 923 magnetic bed -> magnetic bead 

From L 1048 on, the font suddenly gets smaller. 

Author Response

Reviewer’s comment: The manuscript is actually much better to read now.

Author’s comment: Thank you for your comment. That's important for us.

Reviewer’s comment: However, I would still have the following minor points to make:

The newly inserted table 2 is good. However, in my opinion it would be even better and above all clearer, if the respective points would be summarized more strongly, so that one does not have 5 individual entries to a-syn (which here, by the way, is again not uniformly a-syn but once a-synuclein).

Author’s comment: According to your comment, Table 2 is rearranged grouping continuously rows with entries a-syn.

Reviewer’s comment: Under the table comes the summary, and there it starts with a-syn. Therefore it would make sense if the first entry in the table itself would be a-syn and then either all entries summarized to a-syn in the table or at least directly below each other.

Author’s comment: According to your comment, we have changed this table being below of a-syn including derived of a-syn and others potential biomarkers.

Reviewer’s comment: The summary below is really good and also linguistically much better. I found a small error nevertheless:

L 372 there the sources slipped between the word On: O[14,24,38][30]o

Author’s comment: According to your comment, we have changed it in this line.

Reviewer’s comment: The Conclusion is also much much better. Small typos:

L 919 there is a space missing between "sizewas".

L 923 magnetic bed -> magnetic bead

From L 1048 on, the font suddenly gets smaller.

Author’s comment: Thank you for your comment. According to these small comments, we have modified in the manuscript.
